# National Survey Assessment of the United States’ Pediatric Residents’ Knowledge, Attitudes, and Practices Regarding Newborn Screening

**DOI:** 10.3390/ijns5010003

**Published:** 2018-12-31

**Authors:** Shipra Bansal, Kannan Kasturi, Vivian L. Chin

**Affiliations:** 1Section of Pediatric Endocrinology at University of Arkansas for Medical Sciences, Little Rock, AR 72202, USA; 2Pediatric Endocrinology, Pediatric Diabetes and Endocrinology Department at the Essentia Health-Duluth Clinic, Duluth, MN 55805, USA; 3Division of Pediatric Endocrinology at SUNY Downstate Medical Center, Brooklyn, NY-11203, USA

**Keywords:** pediatric residents, newborn screening, knowledge, practices, survey, training

## Abstract

A pediatrician’s approach to newborn screening (NBS) impacts patient care. Some physicians have reported not being well prepared to inform families about a positive NBS and recommend further follow-up. The knowledge and approach of categorical pediatric residents (RES) in the United States regarding NBS is not known. They were anonymously surveyed via listserv maintained by American Academy of Pediatrics. A total of 655 responses were analyzed. The mean composite knowledge score (CKS) was 17.7 (SD 1.8), out of maximum 21. Training level (*p* = 0.001) and completing NICU rotation (*p* < 0.001) predicted higher CKS. Most RES agreed that NBS is useful and pediatricians play an important role in the NBS process, however, only 62% were comfortable with counseling. Higher level RES were more likely to follow NBS results in clinic (*p* = 0.0027) and know the contact agency for results (*p* < 0.001). Most RES wanted more NBS training during residency and were not aware of clinical algorithms like ACTion sheets developed by American College of Medical Genetics. We concluded that although RES have sufficient knowledge about NBS, there is a need for earlier RES education on available tools for NBS to enhance their comfort level and improve practices such as educating parents about the NBS process.

## 1. Introduction

Newborn screening (NBS) is the first population-based public screening program for genetic disorders in the United States (US) [1]. The Center for Disease Control lists NBS as one of the greatest public health achievements in this country [2]. The initial screen for phenylketonuria developed by Dr. Robert Guthrie in the 1960s has now been expanded to many more genetic, metabolic and endocrine diseases [3], with recent additions of hearing screen and congenital heart disease screen [4].

NBS relies on systematic follow-up, confirmation of diagnosis, timely intervention, and treatment [5]. Primary care providers (PCPs) such as pediatricians, family medicine physicians, nurse practitioners, and residents play an important part in the NBS process from obtaining NBS after delivery to diagnosis and treatment. Ideally, the relevance of NBS should be discussed with the parents before specimen collection, even during antenatal visits and subsequently reinforced after the baby’s birth [3,6]. The PCPs inform the parents of the newborn about an abnormal screen and explain its significance. They then further arrange for appropriate confirmatory testing, refer to specialist and continue to monitor the patient [3]. Some family physicians agreed that their own knowledge and comfort with their own role in guiding the family about NBS impact their patient’s care and ultimately, the final result [7]. Kemper et al. reported discomfort among family physicians when discussing abnormal screen results [8]. PCPs have reported a lack of confidence when discussing follow-up for abnormal hearing screen as well [9]. In rural areas, one of the challenges for the NBS program is the limited number of subspecialists available to guide PCPs in the diagnosis and management of these rare conditions [10]. In such scenarios, tools such as ACTion sheets or clinical management algorithms for abnormal NBS results developed by the American College of Medical Genetics can be of great benefit [11].

To our knowledge, there are no studies assessing US pediatric residents’ knowledge, attitudes and practices regarding NBS. Given the importance of NBS, we explored gaps in US categorical pediatric residents’ (RES) knowledge and any trends in attitudes and practices that need to be addressed to improve their education. There were four main areas of interest: (1) knowledge about the information needed to send, interpret and analyze NBS; (2) perspectives including comfort level for counseling for positive NBS results; (3) self-reported practices in the settings of continuity clinic, nursery, and neonatal intensive care unit (NICU) as applicable such as discussing the rationale of NBS with parents, informing them of normal results, and following results; (4) awareness of appropriate follow-up for abnormal NBS, ACTion sheets and the contact agency for NBS results.

## 2. Materials and Methods

An anonymous electronic questionnaire link was emailed to approximately 8000 residents listed in the Section on Pediatric Trainees (SOPT) of the American Academy of Pediatrics (AAP). This listserv includes all pediatric trainees with AAP membership that is usually provided by their residency programs, allowing us to send the survey throughout North America. The questionnaire was intended for categorical pediatric residents in the US. Hence, in the final analysis, medical students, Canadian, and non-categorical pediatric trainees and fellows were excluded. The survey would terminate if a respondent met the exclusion criteria. Responding to the questionnaire was assumed as implied consent. A reminder e-mail was sent again two weeks later to increase response rate. Data collection lasted for one month.

### 2.1. Survey Instrument

The survey was subdivided into sections seeking information regarding responders’ demographics followed by questions assessing knowledge, attitudes and practices. The survey instrument was approved by the AAP section executive committee prior to distribution of the questionnaire, (refer to Appendix A for survey questions). The study was exempt from full review by the institutional review board.
Demographics included year of training, sex, race, state of training, type of hospital, whether a parent of US-born child. Data was collected also for the number of NICU and newborn nursery rotations.Twenty-one multiple choice and true/false questions assessed knowledge of information needed to send, interpret and analyze NBS. Each RES was given a composite knowledge score (CKS).To assess attitudes, RES answered 5 statements using responses based on a Likert scale.Questions regarding the self-reported practices of discussing rationale of NBS with parents, following-up results and discussing normal NBS results in three settings (NICU, well baby nursery and continuity clinic) were asked. Practice responses were categorized as frequencies of 76–100%, 51–75%, 26–50%, 1–25%, 0% of the time and never been in this situation.Assessment of NBS processes was assessed by RES awareness of ACTion sheets and how to use them, appropriate protocol for follow-up, knowledge of specific agency to contact for results were assessed. The survey ended with their interest in a follow-up training session on NBS.

### 2.2. Data Analyses

CKS was calculated for each RES based on the number of correct responses out of a maximum 21. Linear regression with backward elimination was used to determine predictors associated with CKS. Potential factors such as training level, type of residency program, completion of nursery rotation, completion of NICU rotation, and being a parent of a US-born child were entered into models. Attitudes were analyzed using frequency tables. Practices and their association with training levels and number of rotations were tested by Jonckheere–Terpstra test. Hochberg’s method was used to adjust *p*-values for multiple testing and alpha of <0.05 was considered statistically significant.

## 3. Results

From 819 responses received, 152 were excluded due to not being a US categorical pediatric resident, allowing for 8.4% (*n* = 667) response rate out of eligible 7924 US categorical pediatric residents (based on National residency match outcomes for the last three years) [12]. A total of 655 surveys were analyzed after 12 were excluded for incomplete responses.

### 3.1. Demographics

Details of the survey respondents included in final analysis are presented in Table 1.

### 3.2. Knowledge

The mean CKS was 17.7 (SD 1.8), ranging 12–21. In regression modeling, training level (*p* = 0.001) and the completion of NICU rotation (*p* < 0.001) were significant independent predictors associated with higher CKS. (Figure 1). The completion of NICU increased CKS by 1.0 point. Adjusting for NICU rotation, PGY2s, PGY3s, and PGY4s have 0.2, 0.7, and 0.6 points higher CKS on average than PGY1s, respectively. The two predictors accounted for 5% of the explained variability in CKS.

### 3.3. Attitudes

Figure 2 shows RES response based on Likert scale with *n* = 643 due to partial response.

### 3.4. Practices

Higher level trainees (PGY3 and PGY4) are more likely to follow-up results in clinic (*p* = 0.0027, Figure 3) compared to lower level trainees (PGY1 and PGY2), with no statistical difference in the NICU setting. Follow-up of results in the newborn nursery was not asked since NBS results are not expected until after nursery discharge, usually 1–3 days after birth.

### 3.5. Awareness of NBS Processes

The RES understanding of the NBS processes improved with level of training although declined with PGY4, which could be related to poor response rate for PGY4. Higher level RES were also more aware of which agency to contact for results (*p* < 0.0001, Figure 4).

However, there was no association between the number of NICU or nursery rotations and practices such as discussing the rationale for performing NBS with parents, and informing parents of normal NBS results. Similarly, there was no association between higher level RES, those with a US-born child, and the practice of involving the parents in the NBS process.

Only 25% RES reported knowledge about ACTion sheets. Almost half were either not sure or not aware of the protocol for follow-up of positive screen. Similarly, 55% RES wanted a further training session and 32% were undecided.

## 4. Discussion

As far as we know, this is the first study assessing pediatric trainees’ knowledge and approach regarding NBS. Its role in identifying life-threatening illnesses and its importance in a child’s life and not just perinatal period is invaluable. A breach in the NBS process can lead to significant morbidity and mortality which could be easily avoided if NBS processes are appropriately followed. RES, as future independent clinicians, are expected to achieve competence in NBS process during training. Therefore, their education in this process is very crucial. In our survey, a majority of RES concur on the relevance of NBS public health program and the importance of the PCPs in the process.

The RES surveyed have excellent knowledge about NBS, especially about the factors that may alter result interpretation. Higher level RES had better CKS, even after adjusting for NICU rotation and nursery rotation. Interestingly, there was no statistical difference in CKS based on the number of NICU, nursery rotations or being a parent of US-born child. This suggests that overall, the experience gained during training is sufficient for adequate knowledge of NBS, irrespective of the number of specific rotations with neonates.

Despite adequate knowledge, a third of RES reported not feeling comfortable counseling and more than half (56%) were not aware of appropriate follow-up for abnormal NBS. Although our study did not explore barriers, our results are consistent with previous studies on PCP’s attitudes regarding NBS. A study by Kemper et al. revealed discomfort among family physicians when discussing abnormal NBS. Their awareness of ACTion sheets was low, similar to our survey [8]. In another study, almost half of pediatricians surveyed in Massachusetts felt less ready to talk to parents about the results and reported paucity of information about metabolic disorders [13]. Similarly, responding pediatricians in Illinois supported further information to understand a positive and negative screen regarding carrier detection to allow for effective counseling and appropriate referral [14]. One can extrapolate that the RES have similar concerns and hence, resources such as ACTion sheets can prove to be useful. They are free and easily accessible and can guide further workup as needed. Also, the Committee of Genetics under AAP has developed NBS “fact sheets” that discuss pathophysiology and individual tests along with their specific positive and negative attributes as screening tests [15]. It would be of immense interest to evaluate if RES comfort level improves with the awareness of these valuable resources.

Another cause for feeling ill-prepared could be due to lack of experience in counseling itself. The elevated stress level among family of an infant with abnormal NBS has been corroborated with studies of families of newborns with abnormal screen for cystic fibrosis, sickle cell disease, and biochemical disorders [16,17]. Close involvement of PCPs, open communication with honest, clear explanations and avoidance of jargon decrease the parental stress levels with enhancement of their understanding of NBS [18]. One study showed that RES used a large number of jargon words such as gene, sweat test, carrier, and sickle cell during a mock encounter with parents [19]. Additional training such as mock encounters with input from experienced faculty may be useful in boosting RES confidence [20,21]. RES can benefit from improved communication skills not just during encounters for abnormal NBS, but for any situation as it is a necessary part of the core competencies acquired during residency.

The reported parental involvement by RES in the NBS process was not statistically different when adjusted for the number of nursery or NICU rotations or being a parent of a US-born child. We expect that RES who as parents went through the NBS process themselves will empathize and inform parents of normal NBS results as well. Although not informing the parents of normal NBS results is not a serious issue in itself, it is an important part of the newborn’s medical history.

In addition, higher level trainees reported significantly higher rates of follow-up of results in clinic but not during NICU rotation compared to lower level RES. The differences in follow-up by training level may be due to improved knowledge and greater experiences of higher-level RES. More studies are needed to explore the disparity between clinical settings (clinic versus NICU) although it could be related to differences in emphasis for NBS follow-up at the well child check-up. There is room for improvement in follow-up rates in clinic as well. This is similar to the finding that 28% of PCPs surveyed all over US do not actively seek results of NBS [22]. Similarly, more senior RES are aware of the state agency to contact for obtaining NBS results than junior RES as expected. Most RES expressed the need for more NBS training during residency and requested a follow-up session as they probably recognize the significance of NBS.

A study limitation is the low response rate despite our best efforts to invite all eligible RES through the AAP listserv. However, as far as we know, this is the only study assessing RES knowledge, attitudes and practices regarding NBS. Also, we acknowledge the invaluable contribution of other health care providers and nursing teams in NBS process which may influence the clinical practice and procedures at the training hospitals for residents. Nevertheless, it is imperative that they learn about this vital tool in a newborn’s life, including available tools for follow-up protocols.

This is a unique study of its kind to assess the US pediatric trainees’ knowledge, beliefs and perspectives regarding NBS which of immense importance in an infant’s life. Appropriate follow-up of the NBS process can be instrumental in saving an infant from the devastating implications of an otherwise fatal condition. Our study reveals that the RES have sufficient understanding about NBS, with higher training level and completion of NICU rotation increasing the knowledge. However, it highlights the need for improvement in follow-up of results in the NICU setting as well as for lower level trainees. This is important as the procedures followed and learnt during training will influence their future practices. It is equally important to provide the future clinicians with available resources regarding NBS like ACTion sheets. We expect that it will enhance their comfort level and empower them for diagnosis and management of the rare conditions especially if faced with limited subspecialist access and allow to educating parents in the NBS process. Further studies are needed to identify specific barriers in RES education in counseling, communication with parents, and follow-up of NBS results.

## Figures and Tables

**Figure 1 IJNS-05-00003-f001:**
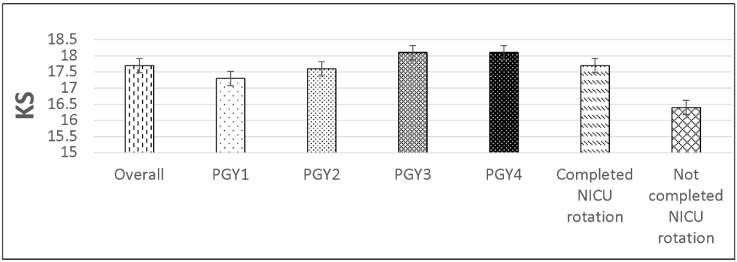
Knowledge score (KS) by level of training (*p* = 0.001) and NICU rotation (*p* < 0.001).

**Figure 2 IJNS-05-00003-f002:**
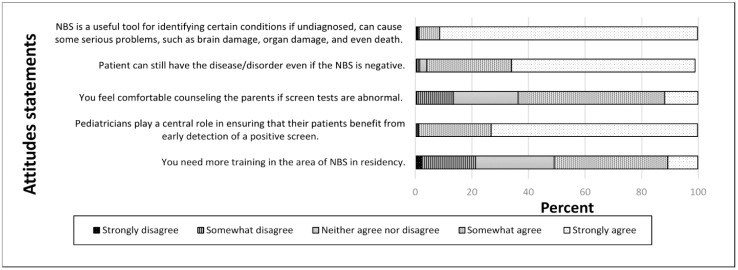
Responses to altitude questions (*n* = 643 due to partial non-response).

**Figure 3 IJNS-05-00003-f003:**
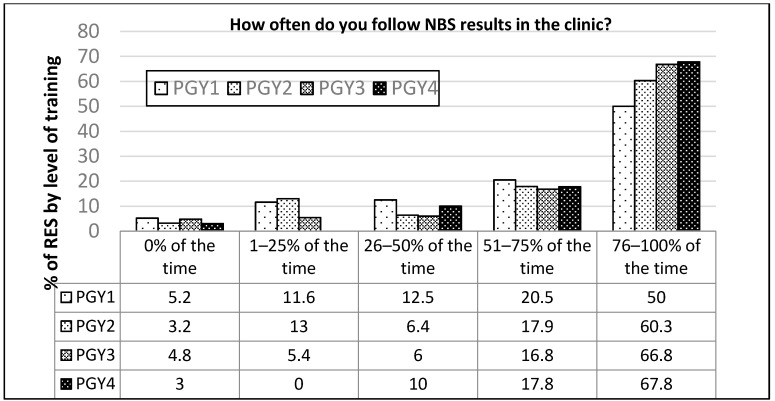
Follow up of NBS results in clinic by training level. Higher level trainees (PGY3 and PGY4) follow results in clinic more often than lower level trainees (PGY1 + PGY2) (*p* = 0.0027).

**Figure 4 IJNS-05-00003-f004:**
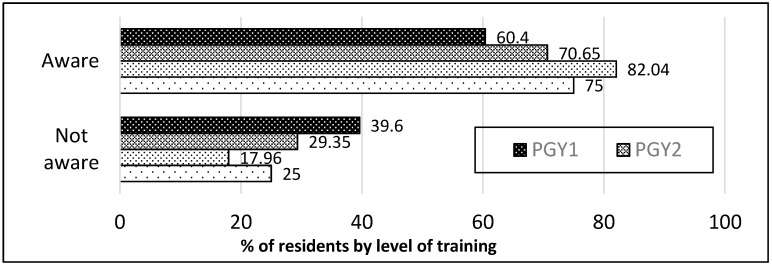
Awareness of agency to contact for NBS results by training level. Higher level RES (PGY3 + PGY4) are most likely to be aware compared to lower level RES (PGY1 + PGY2); (*p* < 0.0001).

**Table 1 IJNS-05-00003-t001:** Demographic characteristics

Gender	*n* (percent)
Male	166 (25%)
Female	489 (75%)
**Level of training**	
First year of training/PGY1	260 (40%)
Second year of training/PGY2	192 (29%)
Third year of training/PGY3	175 (27%)
Chief residents/PGY4	28 (4%)
**Race**	
Non-Hispanic White	459 (70%)
African American	24 (4%)
Asian	98 (15%)
Other/Mixed	47 (7%)
No answer	27 (4%)
**Parent of US-born child**	
Yes	255 (39%)
No	400 (61%)
**Completed at least one NICU rotation**	
Yes	619 (95%)
No	36 (5%)
**Completed at least one Nursery rotation**	
Yes	636 (97%)
No	19(3%)
**Type of residency program**	
University program	352 (54%)
University-affiliated program	228 (35%)
Community hospital program	58 (9%)
Others/No answer	17 (3%)

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
