# Peer review of "National Survey Assessment of the United States’ Pediatric Residents’ Knowledge, Attitudes, and Practices Regarding Newborn Screening"

_2409-515X, 2018, doi:10.3390/ijns5010003_

Round 1

Reviewer 1 Report

Important and relevant topic regarding knowledge of NBS of pediatric residents.  I found figures 3 and 4 difficult to understand; it would be helpful to utilize a side by side bar graph indicating each PGY level separately instead of putting them on top of each other in a single bar.  It would also be helpful to list the actual percentages (for figure 3) or numbers (figure 4) either in the manuscript or supplemental materials. 

The sentence in lines 206-207 needs to be rewritten as it is not clear what the authors mean.

In the listing of authors there seems to be a typo in the superscripts.      

Author Response

Response to Reviewer 2 Comments

Comments

1.     Important and relevant topic regarding knowledge of NBS of pediatric residents.  I found figures 3 and 4 difficult to understand; it would be helpful to utilize a side by side bar graph indicating each PGY level separately instead of putting them on top of each other in a single bar.  It would also be helpful to list the actual percentages (for figure 3) or numbers (figure 4) either in the manuscript or supplemental materials. 

Response: We thank you for their insightful comment. We changed the figures as recommended with percentages of the residents in figures to make them more clear and understandable. Please review the amended manuscript, Figure 3 and Figure 4 on page 5.

2.     The sentence in lines 206-207 needs to be rewritten as it is not clear what the authors mean.

Response: We thank you for this comment for correction. The original manuscript line 206-207 reads as “At the same time, there is important to arm tomorrow’s budding physicians with available resources regarding NBS like ACTion sheets”. In order to better understand the statement, we have changed it as “ It is equally important to provide the future clinicians with available resources regarding NBS like ACTion sheets”. The original statement in the manuscript has been revised in the manuscript, please refer to line 211-12 on page 7.

3.     In the listing of authors there seems to be a typo in the superscripts. 

Response: We thank you for your observation. After careful review, we have amended it as follows and reflects in the updated manuscript:

Shipra Bansal 1, Kannan Kasturi 2 and Vivian L. Chin 3,*

1  Assistant Professor at University of Arkansas for Medical Sciences, Little Rock AR, USA. Email: shiprabansaldr@gmail.com.

2  Pediatric Endocrinologist in Duluth, MN, USA. Email: kannankasturimd@gmail.com

3  Assistant Professor of Pediatrics, SUNY Downstate Medical Center, Brooklyn NY, USA. Email: vivian.chin@downstate.edu

  *   Correspondence: Email: vivian.chin@downstate.edu; Phone: 718-613-8607

4. Moderate English changes were required.

Response: We are grateful for your comments. Please see some of the changes as follows and if there are any further clarifications, we would most certainly appreciate it.

a)    Line 72-75 in the original manuscript “The survey instrument was approved by the AAP section executive committee prior to distribution of the questionnaire. (see Appendix A for survey questions)”. The statement has been changed as “The survey instrument was approved by the AAP section executive committee prior to distribution of the questionnaire, (refer to Appendix A for survey questions).”. Please refer to line 72-75 on page 2.

b)    Line 77-79 on page 2 in the original manuscript “Demographics included year of training, sex, race, state of training, type of hospital, whether a parent of US-born child, number of NICU and newborn nursery rotations were collected.” is now edited as  “Demographics included year of training, sex, race, state of training, type of hospital, whether a parent of US-born child. Data was collected also for the number of NICU and newborn nursery rotations.” Please note on page 2, line 77-79.

Reviewer 2 Report

The manuscript analyzed 655 responses to understand the knowledge of pediatric residents toward newborn screening. The manuscript indicate the need for earlier education to enhance the residents’ knowledge about NBS to improve the practices about the NBS process. Suggestion: Since the 655 responses represent less than 10% of the total 7924 US categorical pediatric residents. It will be better to compare some demographic information between these 2 populations.

Author Response

 Response to Reviewer 2 Comments

Comments

The manuscript analyzed 655 responses to understand the knowledge of pediatric residents toward newborn screening. The manuscript indicate the need for earlier education to enhance the residents’ knowledge about NBS to improve the practices about the NBS process. Suggestion: Since the 655 responses represent less than 10% of the total 7924 US categorical pediatric residents. It will be better to compare some demographic information between these 2 populations.

Response: We thank you for taking time to review the manuscript and for your insightful comment. At this time, we have been unsuccessful in obtaining the detailed demographics as listed in Table 1 for non- respondents. Based on the data in the archives of National Resident Matching Program® , There are about 7924 categorical residents, suggestive of increased number of non-US graduates successfully getting pediatric residency positions. However, there is no freely accessible data on race or gender, whether have a US born child. As can be expected, completion of nursery and NICU rotations (and the number of rotations) would also vary based on individual resident within the same program (residency location).

We agree, that due to the relatively low response rate, this data is difficult to extrapolate to represent all US residents, this data confirms what is seen at the level of practicing physicians in the United States. Please review lines 157-165 on page 6 in the revised manuscript. References noted as 8 and 13 in the section are as follows:

-          Kemper, A. R.; Uren, R. L.; Moseley, K. L.; Clark, S. J., Primary care physicians' attitudes regarding follow-up care for children with positive newborn screening results. Pediatrics 2006, 118 (5), 1836-1841.

-          Gennaccaro, M.; Waisbren, S.; Marsden, D., The knowledge gap in expanded newborn screening: survey results from paediatricians in Massachusetts. Journal of inherited metabolic disease 2005, 28 (6), 819-824